# Graph-Based Multi-Modal Light-weight Network for Adaptive Edge MRI Tumor Segmentation

## Abstract

Brain tumor segmentation plays a critical role in clinical diagnosis and treatment planning, yet the variability in imaging quality across different MRI scanners presents significant challenges to model generalization. To address this, we propose the Edge Iterative MRI Lesion Localization System (EdgeIMLocSys), which integrates Continuous Learning from Human Feedback to adaptively fine-tune segmentation models based on clinician feedback, thereby enhancing robustness to scanner-specific imaging characteristics. Central to this system is the Graph-based Multi-Modal Interaction Lightweight Network for Brain Tumor Segmentation (GMLN-BTS), which employs a Modality-Aware Adaptive Encoder (M2AE) to extract multi-scale semantic features efficiently, and a Graph-based Multi-Modal Collaborative Interaction Module (G2MCIM) to model complementary cross-modal relationships via graph structures. Additionally, we introduce a novel Voxel Refinement UpSampling Module (VRUM) that synergistically combines linear interpolation and multi-scale transposed convolutions to suppress artifacts while preserving high-frequency details, improving segmentation boundary accuracy. Our proposed GMLN-BTS model achieves state-of-the-art (SOTA) performance on both the BraTS2017 and BraTS2021 datasets among lightweight models with only 4.58 million parameters, representing a 98% reduction compared to mainstream 3D Transformer models, and significantly outperforms existing lightweight approaches. This work demonstrates a synergistic breakthrough in achieving high-accuracy, resource-efficient brain tumor segmentation suitable for deployment in resource-constrained clinical environments.

## 1 Introduction

Brain tumors, as life-threatening neurological disorders, have become a global research priority in medicine due to their high mortality and disability rates Louis et al. (2016). In clinical assessment and diagnosis, automated, precise brain tumor segmentation serves as a core module of intelligent lesion localization and disease diagnosis systems Menze et al. (2014). This technology not only significantly enhances diagnostic efficiency but also provides critical quantitative lesion characterization to inform individualized treatment planning. Owing to its superior soft-tissue resolution, multi-modal imaging capabilities, and non-invasive advantages, Magnetic Resonance Imaging (MRI) has been established as the gold standard for neurological disease diagnosis and lesion localization Filippi & Agosta (2010). However, variations in image quality and measurement results arise when imaging brain structures across different MRI scanners (from diverse manufacturers and models) Shokouhi et al. (2011). Concurrently, discrepancies exist among different MRI machines in measuring brain volume and cortical thickness, which may stem from device-related factors affecting imaging artifacts and image quality Le Bihan et al. (2006). Furthermore, MRI field strength exerts a certain influence on susceptibility artifacts Farahani et al. (1990). For MRI-based intelligent lesion localization systems, imaging quality plays a pivotal role in the system's ability to accurately segment and localize lesions. Therefore, enhancing the model's adaptability to imaging from diverse MRI equipment is paramount. To this end, we integrate the Continuous Learning from Human Feedback into our model-equipped devices, proposing an innovative system framework: The Edge Iterative MRI Lesion Localization System (EdgeIMLocSys). Simultaneously, addressing the issues

of existing multi-modal brain tumor segmentation models being either over-parameterized or underperforming lightweight models, we propose a Graph-based Multi-Modal Interaction Lightweight Network for Brain Tumor Segmentation (GMLN-BTS).

Brain tumor segmentation diagnosis typically employs multiple Magnetic Resonance Imaging (MRI) modalities to identify tumor regions, such as Fluid-Attenuated Inversion Recovery (FLAIR), contrast-enhanced T1-weighted (T1c), T1-weighted (T1), and T2-weighted (T2) Menze et al. (2014), and each modality provides distinct structural and pathological contrast in the brain Havaei et al. (2017). and different modalities contribute variably to the identification of distinct tumor subregions: FLAIR is more sensitive to background information, T1ce is more sensitive to necrosis, non-enhancing tumor core (NCR/NET), and gadolinium-enhancing tumor (ET), while FLAIR and T2 are more sensitive to peritumoral edema (ED). This characteristic indicates that different modalities provide complementary information/features for tumor region segmentation. Building upon this, we designed a Graph-based Multi-Modal Collaborative Interaction Module (G2MCIM), which employs graph node interactions and graph edge relationship modeling to achieve interactive modeling and enhancement of complementary features across modalities.

Precise reconstruction of the three-dimensional structure of brain tumors and their sub-regions is crucial in brain tumor segmentation tasks, which highly depend on the decoder's ability to effectively upsample low-resolution feature maps Milletari et al. (2016). Commonly used upsampling operations, such as linear interpolation and transposed convolution, exhibit complementary advantages. Linear interpolation offers good stability; however, it lacks learnable parameters and relies solely on mathematical weighted averaging of local pixel values. Although free from checkerboard artifacts, its inherent smoothing operation on input and output features results in significant low-frequency blurring, which can compromise the recovery of high-frequency details (e.g., tumor boundaries, fine textures). In contrast, transposed convolution employs learnable kernels capable of learning and restoring high-frequency details beneficial for segmentation, such as edges and textures. Nevertheless, it may introduce checkerboard artifacts due to zero-padding and overlapping, potentially undermining the smoothness and accuracy of segmentation boundaries Odena et al. (2016). To leverage the complementary strengths of both linear interpolation and transposed convolution, this paper introduces a novel Voxel Refinement UpSampling Module (VRUM). By synergistically leveraging the stability of interpolation and the detail-restoration capability of transposed convolution, VRUM effectively suppresses artifacts while enhancing high-frequency feature representation.

In conclusion, our contributions are as follows: 1) The Edge Iterative MRI Lesion Localization System (EdgeIMLocSys) is proposed. It integrates the concept of Continuous Learning from Human Feedback, periodically fine-tuning the model based on clinician feedback on segmentation results to enhance its adaptability to specific MRI scanners. 2) A Modality-Aware Adaptive Encoder (M2AE) is introduced. It incorporates 3D Inception blocks to expand multi-scale perception and utilizes Group Normalization (GroupNorm) to stabilize single-modality feature distributions. Concurrently, the output channels are set to 16, striking a balance between feature extraction capability and GPU memory efficiency. 3) A Graph-based Multi-Modal Collaborative Interaction Module (G2MCIM) is presented. This module constructs a modality relationship graph (T1/T1ce/T2/Flair) to learn cross-modal feature interactions via adaptive weights, explicitly modeling the sensitivity differences of different modalities to tumor sub-regions using a graph structure. 4) The Voxel Refinement UpSampling Module (VRUM) is proposed. It aggregates linear interpolation and multi-scale transposed convolution operations to eliminate artifacts while preserving high-frequency information (e.g., edge textures), thereby improving segmentation boundary accuracy. 5) The proposed GMLN-BTS model, utilizing only 4.58M parameters (a 98% reduction compared to nnFormer), achieves state-of-the-art (SOTA) performance on both BraTS2017 and BraTS2021 datasets among lightweight models. This performance significantly surpasses lightweight models such as SegFormer3D and SuperLightNet, and approaches that of the 150M-parameter nnFormer model, demonstrating a synergistic breakthrough in lightweight design and high accuracy.

## 2 RELATED WORK

### 2.1 MULTIMODAL BRAIN TUMOR SEGMENTATION

Multimodal brain tumor segmentation represents a significant research direction in the field of medical image analysis. In recent years, deep learning techniques have achieved remarkable progress in

this domain, giving rise to various mainstream model architectures. These primarily include Convolutional Neural Networks (CNNs), Transformers, state space models (e.g., Mamba Gu & Dao (2023)), and their diverse hybrid variants. CNNs, leveraging their powerful capability for local feature extraction, dominated early research. For instance, 3D UNet Agrawal et al. (2022) effectively captured hierarchical image features through its encoder-decoder structure and skip connections. However, relying solely on simple modality concatenation and implicit network learning for inter-modal relationships may inadequately capture the complementary information present across different modalities. Transformers possess a strong capacity for global context modeling, which is beneficial for understanding the overall structure of tumors and their relationships with surrounding tissues. Nevertheless, their high computational complexity poses challenges for deployment in clinical settings (e.g., TMFormer Zhang et al. (2024)). Mamba, owing to its linear complexity and global modeling capability, demonstrates considerable potential in the field of image segmentation. However, research on Mamba-based multimodal fusion and segmentation is still in its nascent stage and warrants further in-depth investigation. Hybrid models that combine the local feature extraction prowess of CNNs with the long-range dependency modeling capabilities of Mamba/Transformers (e.g., mmFormer Zhang et al. (2022), MedSegMamba Cao et al. (2024)) often struggle to effectively interact complementary information from different modalities in practical studies and exhibit a certain degree of computational redundancy.

## 2.2 Multimodal Interaction of Brain Tumor Characteristic Information

Current multimodal brain tumor feature interaction mechanisms are primarily achieved through attention weighting, Transformer architectures, and customized modules. Among attention-based methods, the Cross-Modal Attention Fusion (CMAF) module proposed in CMAF-Net Sun et al. (2024) facilitates feature interaction between modalities by learning cross-modal attention weights and achieves adaptive fusion with noise suppression capability. However, its attention mechanism design incurs substantial computational overhead. The DiffBTS Nie et al. (2025) approach incorporates attention-weighted features while leveraging contextual constraints to ensure semantic and spatial consistency. Nevertheless, this method exhibits heavy reliance on contextual constraints—inadequate constraints may introduce erroneous biases. In Transformer-based approaches, MicFormer Fan et al. (2024) employs cross-modal Transformer blocks to integrate long-range dependencies across modalities, yet suffers from high computational complexity. Its dual-stream architecture additionally increases communication complexity. The Mamba architecture effectively models long-range dependencies with significantly lower computational complexity than Transformers. For instance, the Learnable Sequenced State-Space Model (LS3M) proposed by Zhang et al. (2025) achieves efficient long-range dependency modeling through dynamic reordering of modal sequences. However, the dynamic reordering mechanism may induce additional computational costs. ACMINet Zhuang et al. (2022) introduces a cross-modal feature interaction module enabling adaptive and efficient feature fusion/refinement, but its intricate design also results in high computational complexity. These methods universally face challenges of elevated computational complexity, which may trigger explosive memory consumption during practical deployment.

## 2.3 Lightweight Multimodal Brain Tumor Segmentation Model

To enable deployment in resource-constrained clinical settings, a series of lightweight multimodal brain tumor segmentation models has emerged. LATUP-Net Alwadee et al. (2025) enhances multiscale feature extraction and small-target segmentation capabilities through parallel convolutions and attention mechanisms. However, excessive reliance on attention mechanisms may result in overemphasis on local features while neglecting complementary inter-modal semantic information. LIU-Net Li et al. (2024) incorporates Inception modules to extract multi-scale features, yet its exclusive use of parallel convolutional kernels with sizes 1, 3, and 5 potentially constrains the modeling capacity for complex features. LR-Net Zhang et al. (2021) employs shift convolutions and Roberts edge enhancement to improve small-target segmentation, whereas spatial shift operations may lead to insufficient feature capture for marginal small targets. SuperLightNet Yu et al. (2025) utilizes a Random Multi-View Drop Encoder and Learnable Residual Skip Decoder to reduce computational load, though such lightweight designs may compromise feature extraction richness. While SegFormer3D Perera et al. (2024) achieves balanced efficiency and accuracy in medical image segmentation through lightweight design and multi-scale attention mechanisms, its strategy of fusing multimodal features via direct concatenation before feeding into a lightweight Transformer may fail

to preserve cross-modal complementary information critical for segmentation. Furthermore, most existing models directly concatenate multimodal features before training, which inherently limits inter-modal interactive modeling capabilities.

# 3 THE PROPOSED METHOD

## 3.1 THE EDGE ITERATIVE MRI LESION LOCALIZATION SYSTEM (EDGEIMLOCSYS)

The Edge Iterative MRI Lesion Localization System (EdgeIM-LocSys) proposed herein comprises three core components: an MRI scanner, an intelligent terminal device, and an internal pre-trained model. The system operates under the supervision of radiologists. The workflow is as follows: After the MRI scanner acquires a scanning sequence of a patient's brain, the

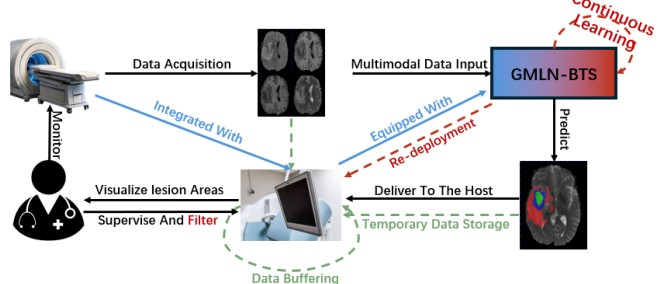

Figure 1: The Edge Iterative MRI Lesion Localization System (EdgeIMLocSys) framework diagram.

resulting images are uploaded to our designed Graph-based Multi-Modal Interaction Lightweight Network for Brain Tumor Segmentation (GMLN-BTS). This model generates a predicted segmentation map, which is then displayed on the screen of the intelligent terminal device. Subsequently, physicians evaluate and filter the segmentation quality, assigning each case a rating categorized as either "Adequate Segmentation" or "Inadequate Segmentation". For each evaluated image, the system stores the original multimodal images alongside the corresponding predicted segmentation result locally. The model undergoes periodic fine-tuning—either weekly or monthly—using the accumulated new data stored locally. This fine-tuning aims to enhance the model's adaptability to the specific imaging characteristics of the individual MRI scanner, thereby facilitating more precise segmentation. Implementing this system relies on a robust foundation of pre-training utilizing extensive MRI brain tumor segmentation data. However, the scale of the BraTS 2017 dataset is inherently limited (i.e., insufficient sample size). Consequently, if large-scale, open-source, multimodal brain tumor segmentation datasets become available in the future, they can be leveraged for both model training and system deployment.

## 3.2 GRAPH-BASED MULTI-MODAL INTERACTION LIGHTWEIGHT NETWORK FOR BRAIN TUMOR SEGMENTATION (GMLN-BTS)

This paper proposes a Graph-based Multi-Modal Interaction Lightweight Network for Brain Tumor Segmentation (GMLN-BTS) (as illustrated in Figure 2). The model first incorporates a Modality-Aware Adaptive Encoder to achieve multi-scale perceptual encoding of information from different modalities. Subsequently, the encoded multi-modal features are fed into the Graph-based Multi-Modal Collaborative Interaction Module (G2MCIM) to facilitate feature interaction across modalities via graph structures. Furthermore, a lightweight Transformer architecture is employed to capture and model global semantic information within the fused features. To enhance model performance during the upsampling stage, a Voxel Refinement UpSampling Module (VRUM) is introduced, which leverages the combined advantages of linear interpolation and transposed convolution to achieve pixel-level enhanced upsampling.

## 3.3 MODALITY-AWARE ADAPTIVE ENCODER (M2AE)

Within the Modality-Aware Adaptive Encoder (M2AE), we introduce a 3D Inception Block (implemented by extending the Inception V1 architecture to 3D space) for semantic-aware modeling of

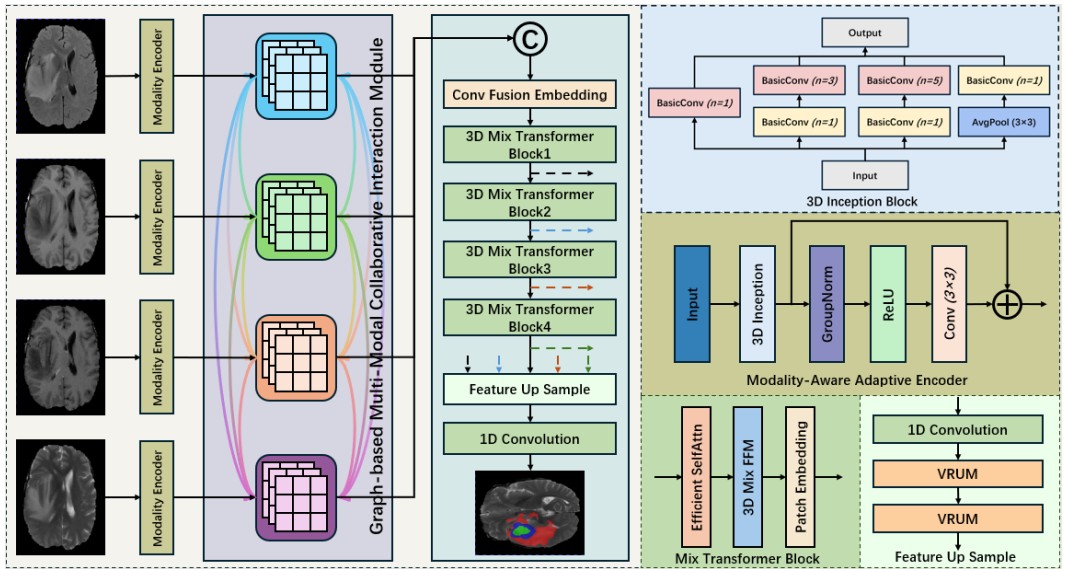

Figure 2: Architectural diagram of the proposed Graph-based Multi-Modal Interaction Lightweight Network for Brain Tumor Segmentation (GMLN-BTS)

multi-scale features in individual modalities:

$$
\begin{aligned}
Y1 &= \mathrm{BasicConv}_{n=1}^{C_{out}/4}(X), \\
Y2 &= \mathrm{BasicConv}_{n=3}^{C_{out}/4}(\mathrm{BasicConv}_{n=1}^{C_r}(X)), \\
Y3 &= \mathrm{BasicConv}_{n=5}^{C_{out}/4}(\mathrm{BasicConv}_{n=1}^{C_r}(X)), \\
Y4 &= \mathrm{BasicConv}_{n=1}^{C_{out}/4}(\mathrm{AvgPool}_{n=3}(X)), \\
Y_{Modality} &= \mathrm{Concat}(Y_1, Y_2, Y_3, Y_4),
\end{aligned}
\tag{1}
$$

where the input tensor $X \in \mathbb{R}^{B \times C_1 \times D \times H \times W}$. GroupNorm is applied to stabilize feature distributions within individual modalities:

$$
\hat{Y}_{Modality} = \mathrm{GroupNorm}(Y_{Modality}).
\tag{2}
$$

Residual connections and activation functions are incorporated to enhance representational capacity and ensure training stability:

$$
\begin{aligned}
Z &= \mathrm{Conv3D}(\mathrm{ReLU}(\hat{Y}_{\mathrm{Modality}})), \\
Z_{\mathrm{Modality}} &= Z + Y_{\mathrm{Modality}}.
\end{aligned}
\tag{3}
$$

The output features $Z_{Modality} \in \mathbb{R}^{B \times C_2 \times D \times H \times W}$ ($C_2 = 16$ to maximize multi-scale feature extraction while minimizing GPU memory consumption), with $Modality \in \{\mathrm{T1, T1ce, T2, Flair}\}$.

## 3.4 GRAPH-BASED MULTI-MODAL COLLABORATIVE INTERACTION MODULE (G2MCIM)

Our proposed Graph-based Multi-Modal Collaborative Interaction Module (G2MCIM) leverages the graph structure and its edge relationships to model the interaction of complementary features across modalities for enhanced representation. First, features encoded by the Modality-Aware Adaptive Encoder across modalities are concatenated to form the output $Z_{Out} \in \mathbb{R}^{B \times 4 \times C_2 \times D \times H \times W}$:

$$
Z_{Out} = \mathrm{Concat}(Z_{T1}, Z_{T2}, Z_{T1ce}, Z_{Flair}).
\tag{4}
$$

To reduce GPU memory consumption, spatial average pooling is applied to compress spatial semantics and extract channel-wise features:

$$
V = \frac{1}{D \times H \times W} \sum_{d=1}^{D} \sum_{h=1}^{H} \sum_{w=1}^{W} Z_{Out}.
\tag{5}
$$

The output features of the above equations are $V \in \mathbb{R}^{B \times 4 \times C_2}$. Then the cross-modal relationship pairs are constructed:

$$R = \text{Concat}(\text{expand}(V, \text{dim} = 1, \text{copies} = 4), \text{expand}(V^T, \text{dim} = 1, \text{copies} = 4)). \quad (6)$$

The output features of the above equations are $R \in \mathbb{R}^{B \times 4 \times 4 \times 2C_2}$. The modality-specific relation encoding network $\phi_i(\cdot)$ is implemented via bilinear layers as:

$$\phi_i(z) = W_{i2} \cdot \sigma_{\text{LeakyReLU}}(W_{i1} \cdot z + b_{i1}) + b_{i2}. \quad (7)$$

This network is then applied to the output features to generate adaptive relational weights for each modality $i \in \{\text{T1}, \text{T1ce}, \text{T2}, \text{Flair}\}$:

$$A_i = \phi_i(R_{:,i,:,:}), \quad (8)$$

Weight normalization is performed to standardize features and mitigate gradient explosion:

$$S_i = \text{softmax}(A_i) = \frac{\exp(A_{i,j})}{\sum_{k=1}^{4} \exp(A_{i,k})}. \quad (9)$$

The output features of the above equations are $S_i \in \mathbb{R}^{B \times 4 \times C_2}$. Modal features are reshaped as:

$$F = \text{Reshape}(Z_{Out}), \quad (10)$$

with $M = D \times H \times W$ and the output features of the above equations are $F \in \mathbb{R}^{B \times 4 \times C_2 \times M}$. Cross-modal weighted fusion is conducted using relational weights:

$$U_i = \sum_{j=1}^{4} S_{i,j} \odot F_j, \quad (11)$$

where $\odot$ indicates channel-wise multiplication. The output features of the above equations are $U_i \in \mathbb{R}^{B \times C_2 \times M}$. The final output is generated through a residual connection:

$$Y_i = Z_i + \text{Reshape}(U_i, [B, C_2, D, H, W]). \quad (12)$$

The output features of the above equations are $Y_i \in \mathbb{R}^{B \times C_2 \times D \times H \times W}$.

### 3.5 VOXEL REFINEMENT UPSAMPLING MODULE (VRUM)

The proposed Voxel Refinement UpSampling Module (VRUM) synergistically integrates the complementary advantages of linear interpolation and multi-scale transposed convolutions. Linear interpolation provides a stable and artifact-free (yet relatively blurry) feature foundation, whereas multi-scale transposed convolutions offer potential high-frequency details. Subsequently, through designed fusion and refinement layers, the module preserves the advantages of detail while utilizing the interpolation foundation to suppress artifacts, address missing information, and smooth the results. In the linear interpolation branch, a 2× upsampling operation is applied to the input feature $\mathbf{X} \in \mathbb{R}^{B \times C \times \frac{D}{4} \times \frac{H}{4} \times \frac{W}{4}}$ (We take the first Voxel

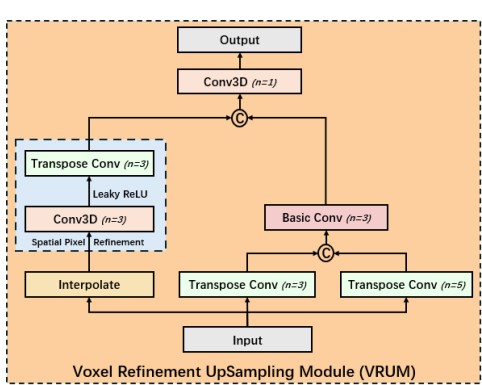

Figure 3: Architecture diagram of the proposed Voxel Refinement UpSampling Module (VRUM).

Refinement UpSampling Module (VRUM) as an example to illustrate its formulation.):

$$\mathbf{X}_{\text{interp}} = \text{TrilinearUpsample}(\mathbf{X}; s = 2). \quad (13)$$

Given that linear interpolation acts as a low-pass filter, we employ a Spatial Pixel Refinement module to perform preliminary refinement on the interpolated features. This module enhances high-frequency information while simultaneously calibrating and reinforcing spatial semantic features. The Spatial Pixel Refinement module comprises a convolutional layer, LeakyReLU activation, and

a transposed convolutional layer. The convolutional layer learns and recovers partially lost local details and edge sharpness:

$$\mathbf{X}_{\text{conv}} = \text{Conv3D}(\mathbf{X}_{\text{interp}}; \quad \mathbf{W}_{\text{conv}}, \text{kernel} = 3, \text{stride} = 1, \text{pad} = 1). \tag{14}$$

The transposed convolutional layer applies region-specific adjustments (e.g., edge sharpening and flat region denoising) to selectively enhance edge features critical for segmentation:

$$\mathbf{X}_{\text{trans}} = \text{ConvTranspose3D}(\text{LeakyReLU}(\mathbf{X}_{\text{conv}}; \alpha = 0.01);$$
$$\mathbf{W}_{\text{trans}}, \text{kernel} = 4, \text{stride} = 2, \text{pad} = 1), \tag{15}$$

where $\mathbf{W}_{\text{conv}}$ and $\mathbf{W}_{\text{trans}}$ denote convolutional and transposed convolutional kernel weights, respectively, with output channels maintained at $C$. This sequential convolution-transposed convolution structure is computationally efficient, enhancing spatial detail quality while remaining deployable on embedded devices. Although this module improves high-frequency information capture in the linear interpolation branch, inherent limitations of linear interpolation still filter out crucial high-frequency components (e.g., fine edges, textures, and details). As such components are vital for segmentation quality, we design a parallel multi-scale transposed convolution branch to augment high-frequency information capture. In the multi-scale transposed convolution branch, we employ transposed convolutions with varying kernel sizes to enhance detail capture while mitigating checkerboard artifacts: Small-kernel transposed convolution (kernel $= 3$) generates sharper outputs with richer details. Its limited receptive field effectively captures local fine-grained variations but increases susceptibility to checkerboard artifacts:

$$\mathbf{X}_{\text{small}} = \text{ConvTranspose3D}(\mathbf{X}; \mathbf{W}_{\text{small}}, \text{kernel} = 3,$$
$$\text{stride} = s, \text{pad} = 1, \text{out\_pad} = s - 1). \tag{16}$$

Large-kernel transposed convolution (kernel $= 5$) produces smoother outputs with improved coherence. It's expanded receptive field models long-range dependencies, significantly suppressing artifacts at the cost of partial detail loss:

$$\mathbf{X}_{\text{large}} = \text{ConvTranspose3D}(\mathbf{X}; \mathbf{W}_{\text{large}}, \text{kernel} = 5,$$
$$\text{stride} = s, \text{pad} = 2, \text{out\_pad} = s - 1). \tag{17}$$

Both branches output features with $C_{\text{mid}} = \lfloor C_{\text{out}}/2 \rfloor$ channels ($\mathbf{W}_{\text{small}}$ and $\mathbf{W}_{\text{large}}$ denote kernel weights). Feature fusion proceeds as:

$$\mathbf{X}_{\text{cat}} = \text{Concat}(\mathbf{X}_{\text{small}}, \mathbf{X}_{\text{large}}),$$
$$\mathbf{X}_{\text{fuse}} = \text{Conv3D}(\mathbf{X}_{\text{cat}}; \mathbf{W}_{\text{fuse}}, \text{kernel} = 3, \text{pad} = 1),$$
$$\mathbf{X}_{\text{bn}} = \text{BatchNorm}(\mathbf{X}_{\text{fuse}}),$$
$$\mathbf{X}_{\text{out}} = \text{ReLU}(\mathbf{X}_{\text{bn}}) \quad (\text{channels} = C_{\text{mid}}). \tag{18}$$

The final upsampled output is obtained through concatenation and convolutional fusion:

$$\mathbf{Y}_{\text{cat}} = \text{Concat}(\mathbf{X}_{\text{trans}}, \mathbf{X}_{\text{out}}),$$
$$\mathbf{Y}_{\text{final}} = \text{Conv3D}(\mathbf{Y}_{\text{cat}}; \mathbf{W}_{\text{fusion}}, \text{kernel} = 1). \tag{19}$$

## 4 EXPERIMENTS

### 4.1 DATASETS

The Multimodal Brain Tumor Segmentation (BraTS) datasets are publicly released, multi-institutional collections of pre-operative, multi-modal MRI scans (T1, T1-contrast enhanced/T1ce, T2, and FLAIR) with expert manual segmentations of tumor sub-regions (enhancing tumor, tumor core, whole tumor), standardized preprocessing (co-registration, skull-stripping, resampling) used to benchmark segmentation algorithms with Dice. BraTS2017, an earlier edition of the challenge, provided about 285 training cases; BraTS2021 is a later, larger release (the 2021 challenge provided roughly 2,000 cases in total) that continues the same multi-modal format and evaluation protocol while supporting the current state of the art in tumor segmentation. Aligned with state-of-the-art methodologies, we employ identical datasets and evaluation protocols to ensure fair and consistent comparisons across all architectures.

### 4.2 IMPLEMENTATION DETAILS

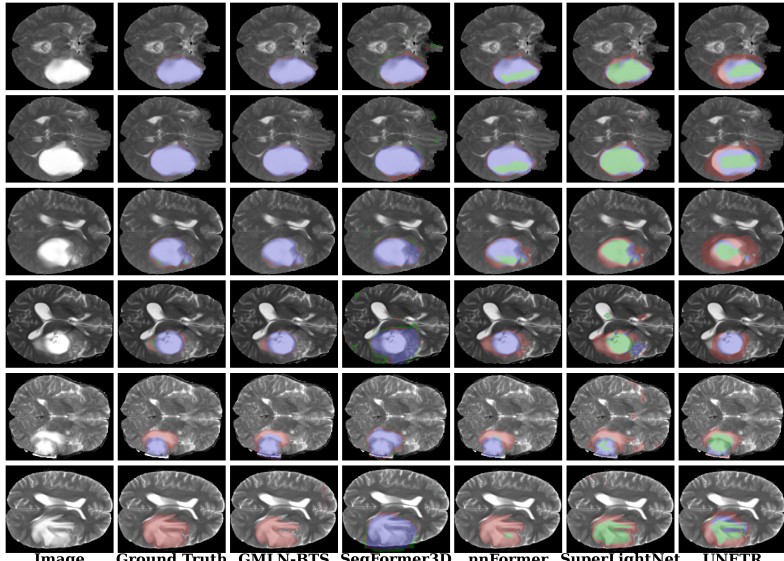

Figure 4: Qualitative comparison of the segmentation performance of different models on the BraTS2017 dataset.

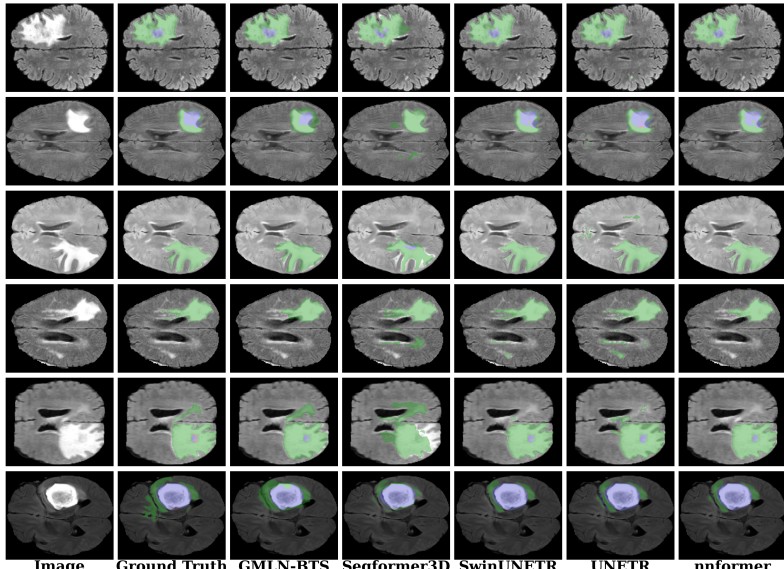

Figure 5: Qualitative comparison of the segmentation performance of different models on the BraTS2021 dataset.

The experimental environment was established on Ubuntu 23.10, utilizing Python 3.12.0 and PyTorch 1.10 accelerated by CUDA 12.1. Hardware configuration consisted of an NVIDIA A800-SXM4-80GB GPU and Intel Xeon Platinum 8462Y+ CPU. Learning rate settings followed Segformer3D: identical scheduling policy with linear warm-up (from $4 \times 10^{-6}$ to $4 \times 10^{-4}$) followed by PolyLR decay. We employed the widely adopted AdamW optimizer with a base learning rate of $3 \times 10^{-5}$. The loss function combined equally weighted

| Method | Params | Avg%↑ | WT↑ | ET↑ | TC↑ |
|---|---|---|---|---|---|
| TransBTS (Wenxuan et al., 2021) | - | 69.6 | 77.9 | 57.4 | 73.5 |
| CoTr (Xie et al., 2021) | 41.9 | 68.3 | 74.6 | 74.6 | 74.8 |
| CoTr w/o CNN Encoder (Xie et al., 2021) | - | 64.4 | 71.2 | 52.3 | 69.8 |
| TransUNet (Chen et al., 2021) | 96.07 | 64.4 | 70.6 | 54.2 | 68.4 |
| SETR MLA (Zheng et al., 2021) | 310.5 | 63.9 | 69.8 | 55.4 | 66.5 |
| SETR PUP (Zheng et al., 2021) | 318.31 | 63.8 | 69.6 | 54.9 | 67.0 |
| SETR NUP (Zheng et al., 2021) | 305.67 | 63.7 | 69.7 | 54.4 | 66.9 |
| nnFormer (Zhou et al., 2021) | 150.5 | 86.4 | 91.3 | 81.8 | 86.0 |
| UNETR (Hatamizadeh et al., 2022) | 92.49 | 71.1 | 78.9 | 58.5 | 76.1 |
| SegFormer3D (Perera et al., 2024) | 4.51 | 82.1 | 89.9 | 74.2 | 82.2 |
| SuperLightNet (Yu et al., 2025) | 2.97 | 77.4 | 84.8 | 66.4 | 80.9 |
| **GMLN-BTS (Ours)** | **4.58** | **85.1** | **90.5** | **81.2** | **83.5** |

Figure 6: Comparison of segmentation performance across different tumor regions and model parameter counts for various models on the BraTS2017 dataset.

Dice loss and cross-entropy to accelerate convergence. All models were trained for 800 epochs with a batch size of 2. All experiments were repeated three times under identical conditions, and the mean Dice coefficient was averaged.

### 4.3 STATE-OF-THE-ART COMPARISON

As quantitatively demonstrated in Figure 6, our proposed GMLN-BTS achieves performance break-throughs with an ultra-lightweight architecture on the BraTS2017 dataset: It attains an average Dice coefficient of 85.1% using merely 4.58M parameters, which is less than 3% of nnFormer and below 1.5% of typical SETR variants. Crucially, it outperforms other similarly-scaled models in key sub-region segmentation—exceeding SegFormer3D by 7.0% in enhancing tumor (ET: 81.2% compared to 74.2%) and by 1.3% in tumor core (TC: 83.5% compared to 82.2%), while also surpassing Su-perLightNet (2.97M parameters) by 2.6% in TC segmentation. Moreover, its whole tumor segmentation performance (WT: 90.5%) closely approaches that of nnFormer (91.3%), which uses 150.5M parameters.The qualitative visualization comparison results of different models are presented in Figure 4.

As quantitatively demonstrated in Figure 7, our proposed GMLN-BTS achieves state-of-the-art (SOTA) performance among lightweight models on the BraTS2021 dataset. With a highly efficient parameter count of only 4.58M, it attains the highest average Dice score of 88.7%, surpassing other compact models like Seg-Former3D (4.51M params / 86.0% Avg) and SuperLightNet (2.97M params / 86.3% Avg).

| Method | Params | Avg%↑ | WT↑ | ET↑ | TC↑ |
|---|---|---|---|---|---|
| nnFormer (Zhou et al., 2021) | 150.5 | 83.7 | 87.9 | 78.4 | 84.7 |
| SwinUNETR (Hatamizadeh et al., 2021) | 62.19 | 89.4 | 92.7 | 88.3 | 90.2 |
| UNETR (Hatamizadeh et al., 2022) | 92.49 | 84.3 | 88.7 | 81.7 | 82.4 |
| SegFormer3D (Perera et al., 2024) | 4.51 | 86.0 | 88.9 | 81.4 | 87.8 |
| SuperLightNet (Yu et al., 2025) | 2.97 | 86.3 | 88.1 | 84.5 | 86.4 |
| **GMLN-BTS (Ours)** | **4.58** | **88.7** | **90.3** | **86.1** | **89.7** |

Figure 7: Comparison of segmentation performance across different tumor regions and model parameter counts for various models on the BraTS2021 dataset.

Notably, GMLN-BTS also achieves top scores across all tumor sub-regions (WT: 90.3%, ET: 86.1%, TC: 89.7%), outperforming not only its lightweight peers but also several larger models. This exceptional performance establishes GMLN-BTS as a new benchmark for accuracy-efficiency trade-offs in medical image segmentation. The qualitative visualization comparison results of different models are presented in Figure 5.

## 5 ABLATION STUDY

### 5.1 THE EFFECTIVENESS OF DIFFERENT COMPONENTS OF THE MODEL ARCHITECTURE

As shown in Figure 8, each component improves segmentation (Mean Dice%). G2MCIM raises performance from 81.9% to 84.2%. Adding M2AE (G2MCIM+M2AE) increases it to 84.7%, highlighting M2AE's role in extracting intra-modal semantics. The full model (ALL) with VRUM reaches 85.1%, 0.4 points above G2MCIM+M2AE, validating the synergy among G2MCIM, M2AE, and VRUM and the superiority of the complete model (GMLN-BTS). This result definitively validates: (1) the synergistic interaction among G2MCIM, M2AE, and VRUM driving comprehensive performance gains; (2) the consistent and significant superiority of the complete model (GMLN-BTS).

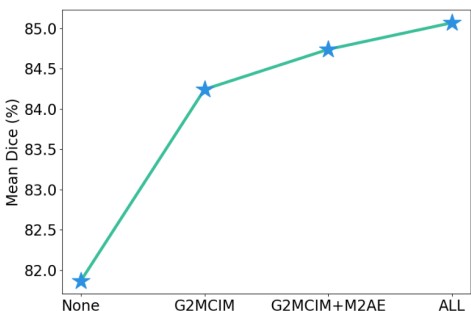

Figure 8: The Impact of Model Components on Performance

## 6 CONCLUSION

The proposed EdgeIMLocSys integrates Continuous Learning from Human Feedback to enhance adaptability to diverse MRI scanners, and the GMLN-BTS model leverages modality-aware encoding, graph-based multi-modal interaction, and refined upsampling to achieve superior segmentation accuracy with minimal parameter overhead. Experimental results on the BraTS2017 and BraTS2021 datasets demonstrate that our lightweight model achieves state-of-the-art (SOTA) performance among lightweight models with only 4.58 million parameters, striking a synergistic balance between efficiency and accuracy, and providing a practical solution for embedded clinical deployment in brain tumor segmentation.

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

# A APPENDIX

## A.1 EVALUATION METRICS

In brain tumor segmentation tasks, the mean Dice similarity coefficient (mean Dice) serves as the core metric for evaluating segmentation performance. This metric objectively assesses segmentation accuracy by quantifying voxel-level spatial overlap between predicted results and expert-annotated ground truth. It is calculated using the Dice coefficient (DSC):

$$\text{DSC} = \frac{2|\text{X} \cap \text{Y}|}{|\text{X}| + |\text{Y}|} \tag{20}$$

where X and Y denote the voxel sets of predicted and ground-truth regions, respectively. Addressing tumor heterogeneity (e.g., irregular morphology and blurred boundaries) and multi-subregion clinical characteristics (enhancing tumor [ET], tumor core [TC], whole tumor [WT]) in BraTS tasks, mean Dice independently computes DSC for ET, TC, and WT subregions and takes their arithmetic mean as the comprehensive score. This design significantly enhances sensitivity to intra-tumoral structural variations—particularly focusing on segmentation accuracy for the highly invasive, small-volume ET region while evaluating overall tumor identification capability, thereby providing a standardized measure of algorithmic robustness in complex clinical scenarios.

## A.2 LLM USE DECLARATION

Large Language Models (ChatGPT) were used exclusively to improve the clarity and fluency of English writing. They were not involved in research ideation, experimental design, data analysis, or interpretation. The authors take full responsibility for all content.

