# OpenReview forum: "Graph-Based Multi-Modal Light-weight Network for Adaptive Edge MRI Tumor Segmentation"
_ICLR.cc/2026/Conference — ICLR 2026 Conference Withdrawn Submission_

### Official Review · Reviewer_sodM · 2025-10-26

**Soundness:** 2
**Presentation:** 2
**Contribution:** 2
**Rating:** 2
**Confidence:** 4

**Summary:**

This paper proposes EdgeIMLocSys, an adaptive edge-based MRI lesion localization system that continuously fine-tunes segmentation models using clinician feedback to handle imaging differences across MRI scanners. At its core is GMLN-BTS, a lightweight, graph-based multimodal network that integrates a Modality-Aware Adaptive Encoder (M2AE) for efficient feature extraction, a Graph-based Multi-Modal Collaborative Interaction Module (G2MCIM) for cross-modal feature fusion, and a Voxel Refinement UpSampling Module (VRUM) to enhance boundary precision by blending interpolation and transposed convolutions. With only 4.58 million parameters, GMLN-BTS achieves state-of-the-art segmentation accuracy on the BraTS2017 and BraTS2021 benchmarks, offering a 98% reduction in model size compared to large Transformer models while maintaining competitive performance—making it highly suitable for real-world, resource-constrained clinical environments.

**Strengths:**

The proposed GMLN-BTS model achieves state-of-the-art segmentation accuracy with only 4.58 million parameters, representing a 98% reduction in size compared to heavy 3D Transformer models. This balance between computational efficiency and accuracy makes it ideal for real-world, resource-limited clinical settings.

**Weaknesses:**

The network design contains too many colors, which affects the overall aesthetic.
I cannot see the details of the graph-based multi-modal interaction; a figure should be added to clarify this.
The datasets used in this paper are outdated.

**Questions:**

N/A

---

### Official Review · Reviewer_NaZf · 2025-10-27

**Soundness:** 3
**Presentation:** 3
**Contribution:** 2
**Rating:** 4
**Confidence:** 4

**Summary:**

This paper proposes GMLN-BTS, which is a lightweight framework for cross-device multimodal brain tumor segmentation, as well as the accompanying EdgeIMLocSys edge-side adaptation pipeline. GMLN-BTS integrates three key components. These components are M2AE for feature encoding, G2MCIM for channel-wise cross-modal graph interaction, and VRUM for artifact-reduced upsampling, all within a model with only 4.58 million parameters. Evaluated on the BraTS 2017 and BraTS 2021 datasets, GMLN-BTS achieves state-of-the-art performance among lightweight methods. Ablation studies validate the contribution of each module, and the manuscript is well-organized with information-rich figures.

**Strengths:**

* The paper demonstrates originality through its use of G2MCIM for per-channel graph-based cross-modal weighting, as well as the integration of trilinear interpolation with multi-scale transposed convolution in VRUM to mitigate checkerboard artifacts. Additionally, EdgeIMLocSys introduces a practical perspective on edge-side adaptation.
* The writing is precise, with module specifications and figures aligned with the formulas provided. Ablation studies, conducted under a transparent three-run setup, show consistent improvements in Dice scores from 81.9 to 85.1.
* The work has notable impact. Specifically, with only 4.58 million parameters, the model achieves state-of-the-art performance among lightweight methods on BraTS2017 and BraTS2021 across the ET, TC, and WT subregions. It also offers a robust efficiency-accuracy trade-off, which makes it well-suited for deployment.

**Weaknesses:**

* Statistical reporting is incomplete, as metrics are presented as means across three runs without accompanying standard deviations, confidence intervals, or significance tests.
* Generalization and system validation are limited. Evaluation is restricted to BraTS2017 and BraTS2021, with no external or cross-center testing; out-of-domain performance remains unquantified. Additionally, EdgeIMLocSys lacks controlled experiments, and key measurements including latency, memory usage, update costs, update frequency, rollback mechanisms, and quality control metrics are absent.
* Reproducibility details are partial. There is a discrepancy between the text and figures regarding VRUM parameters, and this discrepancy specifically appears between the formula and the diagram. Moreover, layer-wise hyperparameters and upsampling details such as output padding are not consolidated, and boundary and artifact-related metrics including Hausdorff95, Surface Dice, and Boundary F-score are missing.

**Questions:**

1. Could you share the precise pseudocode and shape tracing for constructing matrix R (including its expansion, transposition, and permutation operations)? Additionally, please clarify whether the function $\phi_i$ is shared across modalities or specific to each modality, and specify the weight normalization method used in Equation (9).
2. How do you resolve the inconsistency between the text stating kernel = 4, stride = 2 and pad = 1 and the diagram showing $n=3$? Please list the final kernel size, stride, padding and output padding for each branch, specify the magnification factor of each VRUM layer, and explain the alignment between interpolation and multi-scale deconvolution, including layer-wise feature map sizes.
3. Will authors report the mean values with accompanying standard deviations or confidence intervals, and conduct significance tests? Additionally, will authors add boundary and artifact-related metrics such as Hausdorff95, Surface Dice, Boundary F-score or frequency uniformity and provide results from external or cross-center validation? For EdgeIMLocSys, will authors conduct experiments with and without periodic fine-tuning, including measurements of latency, memory usage and update cost?

---

### Official Review · Reviewer_nmtc · 2025-10-30

**Soundness:** 1
**Presentation:** 2
**Contribution:** 1
**Rating:** 2
**Confidence:** 4

**Summary:**

This paper proposes a lightweight multimodal MRI tumor segmentation framework called GMLN-BTS, integrated within an “Edge Iterative MRI Lesion Localization System” for edge-device adaptation via clinician feedback. It combines a 3D Inception-based Modality-Aware Adaptive Encoder, a Graph-based Multi-Modal Collaborative Interaction Module for cross-modal fusion, and a Voxel Refinement UpSampling Module merging interpolation and transposed convolutions. Despite clear writing and reasonable experimental results on BraTS2017/2021, the work lacks genuine novelty, theoretical rigor, and experimental depth: most components are re-combinations of prior CNN/Transformer designs without new insight or verifiable implementation of the “human-feedback” mechanism; thus, it does not meet ICLR’s bar for conceptual contribution.

**Strengths:**

-	The topic (lightweight multimodal segmentation for edge deployment) is practically relevant.

-	Experiments report competitive Dice performance on standard BraTS datasets.

-	The model architecture is simple and reproducible in principle (low parameter count, clear modular design).

**Weaknesses:**

-	All modules (graph-based fusion, Inception encoder, interpolation + transpose upsampling) are well-established and only recombined.

-	No clear graph formulation, adjacency learning scheme, or optimization loss beyond Dice + CE.

-	No comparison of computational latency or edge-device benchmarks to justify “adaptive edge” claims.

-	The “human feedback fine-tuning” aspect is entirely untested.

-	Only BraTS datasets used — no cross-dataset validation (e.g., BraTS2023, ISLES).

-	The narrative exaggerates incremental gains and repeatedly asserts SOTA status without adequate evidence.

**Questions:**

1.	How is the graph edge weight matrix in G2MCIM initialized and updated? Is it learned jointly with segmentation or precomputed?

2.	How exactly is the “continuous learning from human feedback” realized — online updates, reinforcement, or fine-tuning?

3.	What are the actual FLOPs and inference time improvements compared to SegFormer3D on identical hardware?

4.	Can the authors release code or pretrained weights for reproducibility?

---

### Official Review · Reviewer_dUuK · 2025-11-01

**Soundness:** 3
**Presentation:** 3
**Contribution:** 3
**Rating:** 6
**Confidence:** 2

**Summary:**

This paper proposes a lightweight, graph-based, multi-modal network (GMLN-BTS) for brain tumor segmentation from MRI data. The method is designed to be highly parameter-efficient (4.58M parameters) for deployment on edge devices. The core architecture consists of three main components: a Modality-Aware Adaptive Encoder (M2AE) for multi-scale semantic extraction, a Graph-based Multi-Modal Collaborative Interaction Module (G2MCIM) for cross-modal feature interactions, and a Voxel Refinement UpSampling Module (VRUM) to enhance segmentation accuracy by combining interpolation with transposed convolutions. GMLN-BTS achieves state-of-the-art performance on BraTS datasets with significantly fewer parameters compared to existing methods.

**Strengths:**

1. The introduction of a graph-based interaction module (G2MCIM) and adaptive voxel refinement (VRUM) is innovative, effectively leveraging multimodal and cross-modal information for improved segmentation accuracy.

2. Comprehensive evaluations demonstrate robust performance improvements, achieving competitive segmentation results with substantially fewer parameters, highlighting efficiency without sacrificing accuracy.

3. This lightweight and efficient network addresses practical clinical constraints, showing promise for deployment in resource-limited clinical settings by enhancing adaptability to scanner-specific imaging variations.

**Weaknesses:**

1. The generalizability of the proposed model beyond brain tumor segmentation and its adaptability to other multimodal medical imaging tasks are not extensively explored.

2. The reliance on clinician feedback for continuous learning might pose practical implementation challenges in clinical workflows, especially in high-throughput settings.

3. Although efficiency is emphasized, details on computational speed, inference latency, and resource usage during practical deployment scenarios are not extensively addressed, potentially limiting practical insights.

**Questions:**

1. Could the authors elaborate on the practical implementation challenges of the Edge Iterative MRI Lesion Localization System (EdgeIMLocSys), particularly regarding the frequency and volume of clinician feedback required?

2. How does the model perform on other anatomical regions or lesions beyond brain tumors, and have there been attempts to generalize or validate the architecture in other medical imaging tasks?

3. Could the authors provide details on computational performance metrics (inference speed, memory usage, etc.) compared to existing lightweight models to further highlight practical deployment advantages?

---

### Note · Authors · 2025-11-25

I have read and agree with the venue's withdrawal policy on behalf of myself and my co-authors.